# Guided Plasma Application in Dentistry—An Alternative to Antibiotic Therapy

**DOI:** 10.3390/antibiotics13080735

**Published:** 2024-08-05

**Authors:** Tara Gross, Loic Alain Ledernez, Laurent Birrer, Michael Eckhard Bergmann, Markus Jörg Altenburger

**Affiliations:** 1Department of Operative Dentistry and Periodontology, Center for Dental Medicine, Medical Center–University of Freiburg, Faculty of Medicine, Albert-Ludwigs-University of Freiburg, Hugstetter Straße 55, 79106 Freiburg, Germany; tara.gross@uniklinik-freiburg.de (T.G.); laurent.birrer@uniklinik-freiburg.de (L.B.); 2Center for Tissue Replacement, Regeneration & Neogenesis (GERN), Department of Operative Dentistry and Periodontology, Medical Center, Faculty of Medicine, University of Freiburg, 79108 Freiburg, Germany; 3Laboratory for Sensors, Department of Microsystems Engineering (IMTEK), University of Freiburg, 79110 Freiburg, Germany; ledernez@imtek.de (L.A.L.); bergmann@imtek.de (M.E.B.)

**Keywords:** cold atmospheric plasma, non-thermal plasma, antibacterial agents, antimicrobial, antibiotic resistance, dentistry

## Abstract

Cold atmospheric plasma (CAP) is a promising alternative to antibiotics and chemical substances in dentistry that can reduce the risk of unwanted side effects and bacterial resistance. AmbiJet is a device that can ignite and deliver plasma directly to the site of action for maximum effectiveness. The aim of the study was to investigate its antimicrobial efficacy and the possible development of bacterial resistance. The antimicrobial effect of the plasma was tested under aerobic and anaerobic conditions on bacteria (five aerobic, three anaerobic (Gram +/−)) that are relevant in dentistry. The application times varied from 1 to 7 min. Possible bacterial resistance was evaluated by repeated plasma applications (10 times in 50 days). A possible increase in temperature was measured. Plasma effectively killed 10^6^ seeded aerobic and anaerobic bacteria after an application time of 1 min per 10 mm^2^. Neither the development of resistance nor an increase in temperature above 40 °C was observed, so patient discomfort can be ruled out. The plasma treatment proved to be effective under anaerobic conditions, so the influence of ROS can be questioned. Our results show that AmbiJet efficiently eliminates pathogenic oral bacteria. Therefore, it can be advocated for clinical therapeutic use.

## 1. Introduction

The peculiarities of the oral cavity in general pose a challenge for the elimination of pathogenic germs. (i) The complex ecosystem of the oral cavity harbors a variety of heterogeneous microorganisms, in fact up to 1000 different species [1,2], such as bacteria, fungi, viruses, archaea and protozoa [3]. (ii) The oral cavity provides diverse niches that are difficult to access by the patients and therefore favor undisturbed proliferation and formation of complex three-dimensional (3D) biofilm. (iii) The oral mucosa is particularly sensitive to chemical substances, e.g., hexetidine-, chlorhexidine- or essential oil-containing mouthwashes like Listerine, which can lead to oral ulcers [4], reduced cell proliferation and adhesion [5]. Thus, the elimination of pathogenic germs must be ensured while considering the preservation of healthy cells. In healthy organisms, the potential pathogenicity of bacteria is balanced out by the host’s immune system [6]. However, a shift in the balance, e.g., reduced immune defense or proliferation of pathogenic germs, can contribute to the development of biofilm and thus various oral diseases, e.g., caries or periodontitis [3]. In an organized oral biofilm, microorganisms coexist in an extracellular composition of mostly environmental deoxyribonucleic acid (eDNA), proteins and polysaccharides [7]. As a component of a biofilm, bacteria are known to have different gene expression, transcription and translation compared to their planktonic form, which enables them to grow in pathogenicity. Additionally, bacteria protected by biofilm layers benefit from the blockage to antibiotics and host immune cells and, thus, are able to develop 250 to 1000 times higher resistance to antibiotics [7,8,9]. Especially in dentistry, where antibiotics should only be used for severe infections and extra-oral swellings or when systemic complications are expected, an unnecessarily high number of antibiotics are still regularly given to patients [10,11,12] during routine procedures [13]. For example, studies from the UK as well as the US examining antibiotic administration in dental practices found that as many as about 80% of antibiotic applications were administered without the correct indication [14,15].

Hence, since the discovery of antibiotics in 1928, the risk of bacterial resistance has increased uncontrollably and has become a major global health concern [7]. Without alternative treatments, the annual global death toll from antibiotic resistance is projected to increase from 700,000 to 10 million by 2050 [16,17,18]. Therefore, new therapeutic antimicrobial approaches, especially in dentistry, are urgently required in which antibiotics can be avoided. One promising prospect is the application of plasma.

Plasma is an electrically neutral, or quasi-neutral, ionized gas, also referred to as the fourth state of matter. It contains highly reactive oxygen and nitrogen species (RONS), excited molecules, charged particles, chemically reactive neutral particles, free radicals and ultraviolet (UV) radiation [19,20]. Several studies have already proven the effectiveness and antimicrobial properties of plasma [21,22,23,24] depending on the plasma source and parameter settings [25].

Plasma is already being used regularly in dermatology to improve the healing of chronically infected wounds by reducing the topical bacterial load and promoting healing mechanisms [26,27,28,29]. Here, even beneficial properties on healthy surrounding tissue were reported [21]. However, plasma has a limited disinfecting efficacy as it is ignited by a dielectric barrier discharge (DBD) due to a low energy input. An additional limitation comes with the decreasing efficacy with increasing working distance between the plasma source and the target [30].

Recent studies qualify plasma jet applications as especially efficient in deactivating biofilms. Through the production of small, handy devices, e.g., plasma jets, oral application seems to be within reach [6]. Fields of dental application that are already under investigation, such as sterilization [31], carious lesion control [32] and improvement of adhesive systems [33], as well as tooth whitening [34], root canal disinfection [25] and periimplantitis therapy [35], have so far delivered consistently positive results [36]. However, as mentioned above, it is also required for plasma jets to keep the distance to the plasma source as small as possible. As the distance between the device and the inflammation process increases, the power of the plasma device must also be increased to achieve the same efficacy. Unfortunately, this leads to a rise in temperature and possible damage of cells in the treated tissue [36]. The near-tissue ignition of conventional plasma jets is still not possible.

In general, plasma can be classified into high-temperature, thermal and non-thermal groups. Unlike high-temperature and thermal plasma, in non-thermal plasma, i.e., cold atmospheric plasma (CAP), heavy particles are at room temperature, which enables it to be used at <40 °C [6]. In medicine, temperature control is one of the most important factors in preserving surrounding cells and tissue [36].

The disinfecting mechanism of CAP is not fully established yet. However, different approaches have been described. For one, biological pathways, like DNA damage, lipid peroxidation, protein modulation, induced apoptosis, cell leakage, electrostatic rupture of the membrane, etc., were proposed to explain the disinfecting effect of plasmas [37,38]. Some studies claim that those pathways are triggered by one or more components constituting the plasma. Electricity (in forms of electrical field, ions and electrons) may play a dominant role in deactivating/eliminating Gram-negative bacteria, whereas reactive species (oxygen- or nitrogen-based species, so-called ROS and RNS) may be decisive criterion for eliminating Gram-positive bacteria [20,39,40,41,42].

To improve the mechanism of CAP, a device with a disruptive electrode arrangement for plasma ignition was developed for our study. In contrast to other devices, this arrangement uses the target surface as an electrode so that CAP is ignited directly and directed into the treated area. The effectiveness of the plasma device, called AmbiJet, was tested under aerobic and anaerobic conditions and can be explained by two mechanisms. Firstly, the effect caused by the helium plasma itself, and secondly, the effect of ROS created indirectly by the helium plasma reacting with the oxygen-containing atmosphere. The amount of ROS diffusing away from the immediate plasma jet outskirt increases over time, which results in an increase in the width of the bacteria-free area as a function of time under aerobic conditions (Figure 1, Left). Under anaerobic conditions, on the other hand, no ROS is created, and the results show the direct plasma effectiveness only (Figure 1, right).

The aim of this study was to present and evaluate the efficacy and safety of a device AmbiJet, which can generate a helium (He) plasma jet on the infected tissue. The elimination of aerobic and anaerobic bacteria that are relevant in dentistry and their potential formation of bacterial resistance after plasma application should be determined. Also, the temperature increase during plasma application was investigated.

Overall, these examinations should contribute to the daily clinical use of plasma in dentistry and to the search for an alternative to antibiotic administration to combat bacterial resistance and maintain antibiotic efficacy for severe, possibly life-threatening, diseases.

## 2. Results

### 2.1. Plasma Effect on Aerobic Bacterial Cultures

In a first setup, plasma was applied to various aerobic bacteria, such as Gram-negative bacteria *E. coli* and *P. aeruginosa*, as well as Gram-positive bacteria, like *S. aureus*, *S. mutans* and *E. faecalis*, which generally showed efficacy, i.e., led to elimination of the selected monocultured bacterial forms in all samples (examples: Figure 2).

For all bacteria, a dependency of treatment time and line width could be observed. However, the minimum width of the bacteria-free area was at least equal to the width of the inner diameter of the nozzle in all samples. For one minute of treatment, the bacteria-free line was wider on samples with *P. aeruginosa* and narrower with *S. mutans* and *E. faecalis* (Figure 2b). In general, plasma appeared to be slightly more effective with Gram-negative than Gram-positive bacteria, though the difference was not significant. As shown in Figure 2a, the bacteria-free line for the 1-minute treatment is about 2 mm wide for *E. coli*, which is much larger than the inner diameter of the nozzle (0.8 mm). The width of the bacteria-free line increases with treatment time (Figure 2b and Figure 3). The application of plasma on *E. coli* resulted in complete eradication of all bacterial species up to a bacterial load of 6 log10 of a surface area of 40 mm^2^ (10 mm length × 4 mm width) after 3 min.

In general, plasma appeared to be most effective with *P. aeruginosa* and least effective with *E. faecalis* and *E. coli*, whereas the elimination efficacy decreased with time with *E. coli* and increased with *E. faecalis*. As a side observation, no negative effects appeared on the surrounding agar.

#### Correlation between Efficacy and Treatment Time under Aerobic Conditions

For all aerobic bacteria, a strong linear effect could be observed between treatment time and treatment effect (width of the eradicated area) when R^2^ = 0.715.

### 2.2. Plasma Effect on Anaerobic Bacteria

The plasma treatment showed a high bactericidal effect in all samples with the anaerobic bacteria even under anaerobic conditions. Here again, the plasma seemed to be slightly more efficient against Gram-negative than against Gram-positive bacteria, though not significantly (Figure 4).

No time-dependency could be observed with anaerobic bacteria. The width of the bacteria-free line remained consistent for *F. nucleatum* and *A. odontolyticus*. For *A. actinomycetemcomitans,* variations in efficacy could be observed. These were not caused by experimental variations in the distance between the nozzle and the substrate because seven different treatment times were used on each agar plate.

#### Correlation between Efficacy and Treatment Time under Anaerobic Conditions

In contrast to the findings with aerobic bacteria, no linear time dependency in relation to plasma application could be found for anaerobic bacteria. This is explained by the disinfection mechanisms described in Figure 1.

### 2.3. Resistances after Repeated Plasma Application

As presented in Figure 5, plasma treatment was effective with all bacterial species, for all treatment times and over all treatment cycles. The area of efficacy was larger than the outlet of the nozzle in all cases. The width of the bacteria-free agar-agar remained steady over the observation period with slight inconsistences. The development of bacterial resistance would have led to a decrease in the line width over the cycle number, which is not the case. This effect of inconsistences could be seen in groups treated for one minute in cycle 4 to 7 with a decreased efficacy. However, the efficacy increased from cycle 8 on. All in all, plasma efficiency could be reliably demonstrated in the comparison of the 1st and 10th cycles for each tested bacterial culture. No CFUs appeared on the agar gel at any observation time in the efficacy area.

### 2.4. Temperature Measurements

A rapid rise in temperature could be observed on the implant (sensor 1 measuring the implant temperature) in the first minute of both treatment positions A and B (Figure 6). In treatment position A (sensor and treated site on opposite sides of the implant), the temperature reached a plateau of 37.5 °C and a maximum of 39.5 °C after 5 min. In treatment position B (sensor at an angle of 90° to the treated spot), the temperature reached a plateau of 39.5 °C and a maximum of 40 °C after 5 min. In both cases, the temperature of the bone (measured by sensor 2) rose slowly and reached a maximum temperature of 38.5 °C, which is clinically acceptable.

## 3. Discussion

Medical devices are susceptible to bacterial adhesion, a latter biofilm accumulation and formation causing host defenses and possible failure of the medical device [43]. Known causes of such infections are so-called ESKAPE bacteria, such as *Enterococcus faecalis*, *Staphylococcus aureus*, *Klebsiella pneumoniae*, *Acinetobacter baumannii*, *Pseudomonas aeruginosa* and *Enterobacter* spp. [44,45]. This problem also occurs in dentistry among various biofilm-associated dental infections, but especially with dental implants, where the oral microbiome neighbors the implant surface and thus can easily contaminate the implant surface [46]. Several studies have suggested that plasma application can successfully eliminate or inactivate bacterial colonization (overview in [47]), even biofilms [48]; thus, plasma could offer great advantages in dental medicine. Additionally, antibiotic therapy and antimicrobial chemical substances like CHX [49] are being increasingly criticized due to the development of bacterial resistance, which could increase the risk of global health [7], meaning that innovative antibacterial techniques are urgently required.

Unlike antibiotics or other chemical substances, the plasma does not require a specific target to act, so the bacteria have no chance to adapt and find defense mechanisms [50]. Recent studies suggest that CAP can eliminate bacteria such as *Porphyromonas gingivalis* [24], *P. aeruginosa*, *E. coli* and even multi-drug-resistant *S. aureus*, which is particularly difficult to manage with conventional antibiotic therapy [51,52]. In addition, a steady elimination of the oral Gram-negative bacterium *Fusobacterium nucleatum* was observed. This bacterium can communicate between various oral biofilm colonizers and, for example, assist *Porphyromonas gingivalis* in pathogenicity [53], but above all, it is currently criticized for promoting certain types of cancer such as colorectal carcinoma [54].

As far as we are aware, all studies reported in the literature were conducted with the presence of oxygen, whether directly as a gas admixture in the plasma or indirectly from the surrounding air. Thus, for the first time, the present study included an experimental setup with the absence of oxygen. Hence, the findings deliver further clues on how plasmas interact with microorganisms.

The present study confirms the high disinfecting efficiency of the AmbiJet CAP device on aerobic, anaerobic, Gram-positive and Gram-negative bacteria. The treatment led to a complete eradication of the bacterial load of up to 6 log10 in the treated region within 1 min. The bacteria-free zone was larger than the nozzle of the device because it is partly related to the width of the plasma jet, which is, in turn, related to the diameter of the gas canal that helium is forming as it flows out of the nozzle. Inconsistences in efficacy with *A. actinomycetemcomitans* can be associated with different growing times and inaccuracies in agar plating. Decisive factors influencing the result of plasma efficacy appear to be the type of bacteria as well as plasma exposure time [24].

The effect of plasma on the membrane integrity has also been observed on Gram-negative and -positive bacteria [55]. In this study, CAP from the AmbiJet device is in direct contact with the bacteria, and electricity-based (physical) processes dominate, leading to faster processes and less difference between Gram-positive and Gram-negative bacteria, as is the case in the present investigation for short treatment times. The process can be led by an electrical field [42] and/or charged particles such as electrons; UV radiation does not seem to play a significant role [56,57].

For aerobic bacteria, a linear dependency could be observed between treatment time and treatment effect (width of the eradicated area), which is in good agreement with the literature [58]. In contrast, the results of the present investigation showed no such effect for anaerobic bacteria. The general different behavior between aerobic and anaerobic bacteria strains may not be rooted in the kind of bacteria but in the conditions in which the plasma develops. Reactive oxygen species (ROS such as O_2_^−^ or OOH^−^) are believed to play a role in the decontamination efficacy [42]. Others have reported to observe the development of bacterial tolerance to CAP when the effect of the plasma process relies on intracellular processes mediated by ROS [59,60,61,62]. In this study, under aerobic conditions, the energy within the helium plasma is partly transferred to surrounding oxygen molecules creating ROS around the plasma jet, with a decreasing density with increasing distance from the plasma jet. ROS may therefore propagate the bactericidal effect further away from the plasma (in the so-called “afterglow”), though with a lower efficacy. This would explain the observed steady increase in the line width with the treatment time under aerobic conditions (with aerobic bacteria) and the constant width of the bacteria-free line under anaerobic conditions (with anaerobic bacteria). The influence of oxygen rather than the bacteria type is confirmed by the experiment of Mahasneh et al. who observed an increase in the inhibition zone with the exposure time for helium plasma for anaerobic *Porphyromonas gingivalis* with a plasma treatment performed under aerobic conditions and explained by the presence of ROS [63]. This indicates two things. Firstly, that RNS play no role in the plasma disinfection mechanism in the present study because, otherwise, the bacteria-free line would increase in width under anaerobic conditions in the present study (anaerobic chamber filled with nitrogen). Secondly, that ROS are not necessary, at least when the plasma is in direct contact with the target. Here, ROS may play only a secondary role on the outskirts of the plasma jet in contact with the target (Figure 1, Left). After repeated plasma application in the present study, no development of bacterial resistance could be observed in an observation period of 10 cycles in 50 days, which is corresponding to recent studies investigating bacterial resistance development after stress application [64,65]. One possible explanation is the non-selective elimination by the plasma. In fact, previous studies suggest that plasma application can inhibit bacterial gene transfer by conjugation [59].

In order to provide safe, comfortable and efficient plasma application without overheating the surrounding tissue, the ignition should be insured directly at the site of action. The present study confirms the high efficiency of the presented plasma device AmbiJet, even with an effective area larger than that of the plasma exit point, without temperature increase or visible effects on the adjacent area. The light phenomenon of AmbiJet can be used to visualize the treated surface and might serve to navigate during latter free-hand treatment. Some bacteria such as *E. faecalis* or *S. aureus* carry heat chaperone proteins that help to protect the bacterial proteins from overheating and maintain homeostasis. This means that the bacteria have also developed adaptation mechanisms in this area. In turn, repeated overheating of the application site could lead to bacterial resistance [62]. Therefore, an application < 40 degrees is all the more important.

## 4. Materials and Methods

### 4.1. Preparation of Bacteria and Plasma Treatment

The following bacteria were used for the experiments: aerobic Gram-negative bacteria *Pseudomonas aeruginosa* (ATCC 27853) and *Escherichia coli* (ATCC 25922), aerobic Gram-positive bacteria *Streptococcus mutans* (DSM 20523), *Staphylococcus aureus* (ATCC 25923) and *Enterococcus faecalis* (T9), anaerobic Gram-negative bacteria *Aggregatibacter actinomycetemcomitans* (SUNY1039-9) and *Fusobacterium nucleatum* (DSM20482) and anaerobic Gram-positive bacteria *Actinomyces odontolyticus* (HP-6-13).

The different bacteria strains were cultivated in tryptic soy broth (TSB) medium (Becton Dickinson GmbH, Heidelberg, Germany), applied on Müller–Hinton agar plates (diameter 85 mm, Biomerieux, Nürtingen, Germany) with a glass spatula and stored in an incubation chamber (CellXpert^®^ C170i, Eppendorf, Wesseling-Berzdorf, Germany) for 30 min to immobilize the bacteria on the agar by evaporating the TSB. The bacterial density applied on the agar corresponded to a concentration of approximately 10^6^ bacteria/mL, which was confirmed by dilution series and colony forming unit (CFU) counting using a light microscope (Zeiss Axiovision, Oberkochem, Germany). The bacterial density in relation to multiplication speed of the individual bacteria had been determined in preliminary tests in order to maintain reproducibility and equal experimental conditions. Atmospheric low temperature plasma was ignited using AmbiJet (Freiburger Medizintechnik GmbH, Freiburg, Germany) with a customized applicator (equivalent to the inDrive applicator, Freiburger Medizintechnik GmbH, Freiburg, Germany). The device generates a pulsed AC helium discharge between the nozzle (powered electrode) of the applicator and the target (agar), which is contacted electrically to form the counter-electrode. The electric field between the powered electrode and the counter-electrode guides the plasma towards the target. The nozzle has an inner diameter of 0.8 mm. The helium flow is as low as 0.5 L/min. This low amount of gas can be extracted very easily by the aspiration system of a dental chair and is therefore unproblematic. The temperature of the plasma jet is about 40 °C. To standardize the plasma application, the applicator was mounted in a modified 3D printer (Renkforce, Conrad Electronic SE, Hirschau, Germany) after removing the printer head (Figure 7). The x-, y- and z-axis movement were programmed in g-code using a text editor.

### 4.2. Evaluation of the Efficacy of Cold Atmospheric Plasma on Different Bacterial Strains

Aerobic bacterial strains were plasma-treated in an aerobic facility, while anaerobic bacteria were plasma-treated in the same setup but under anaerobic conditions in a nitrogen-flooded glovebox.

The treated areas were lines of 1 cm in length. The infected agar surfaces were treated for 1, 2, 3, 4, 5, 6 and 7 min. The treatment time was varied by varying the number of passes over the same line (e.g., 1 min means 6 passes over the 1 cm line). The distance from the tip of the nozzle to the agar surfaces amounted between 1 and 2 mm and the nozzle moved with a velocity of 60 mm/min (=1 s/mm). During the treatment, the plasma could be seen touching the agar (Figure 8). Because the plasma could easily be seen, it was used to visualize the treated area and navigate during manual treatment.

After the respective treatment, the agar plates were stored in an incubator unless the colony forming units (CFUs) were already clearly visible and countable. Subsequently, the width of the bacteria-free areas (lines) were photographed and measured (*n* = 10) using the software Fiji (based on ImageJ2) including a ruler for calibration.

### 4.3. Evaluation of Possible Resistance of the Bacteria against the Plasma Treatment

Dealing with bacterial resistance is a major challenge in the healthcare sector and should be limited or at best eliminated. Therefore, AmbiJet was tested for possible development of resistance. Four bacterial strains (*Escherichia coli*, *Enterococcus faecalis*, *Pseudomonas aeruguinosa* and *Staphylococcus aureus*) were cultivated in TSB, plated out on agar plates, plasma-treated, harvested and cultivated again. To breed bacteria that are possibly more resistant to the plasma, an area of 1.5 × 1.5 cm on the agar plate was treated with a higher nozzle speed to stress the bacteria. Based on our preliminary tests and to maintain reproducibility we kept around 30% of the originally seeded bacteria alive. The plates were then stored in an incubator until CFUs were clearly visible. These CFUs were harvested from the agar surface and cultivated in TSB. This breed was plated again on agar plates for the next cycle. At the end of each cycle, the selected bacteria were placed on an agar plate and an efficacy evaluation test was performed as described in the previous section. This selection procedure was repeated 10 times (10 cycles) in accordance with recent studies [64,65] over a period of 50 days.

### 4.4. Statistical Analysis

The data were analyzed and presented in figures and tables using IBM SPSS Statistics 29.0.0.0 (241) (IBM Deutschland GmbH, Ehningen, Germany).

### 4.5. Temperature Measurement

To evaluate a possible temperature increase at the dental implant or alveolar bone during plasma application, an implant (Straumann 4.1 mm 14 mm, SLA active bone level) was inserted into the lower jaw of a pig (Figure 9, Left). Temperature sensors were placed on the implant and in the bone around the implant (Figure 9, Right). The access flap was closed with a continuous seam. The plasma was ignited either in position A (on the opposite side of the implant in regard to the position of the sensor) or in position B (on the side of the implant).

The calibrated pt100 temperature sensors were connected via a MAX31865-Interface to an Arduino Uno R3 (all: Conrad Electronic SE, Hirschau, Germany) to log the temperature before, during and after the treatment. Each setup was placed in a water-bath and the measurement was started once the surrounding agar or jaw had reached 35 °C.

## 5. Conclusions

AmbiJet was designed specifically for application in dentistry. It is very effective, unselective toward the bacteria and does not harm the surrounding tissues. The present study confirms the high efficiency of the AmbiJet plasma device without thermal impact on the adjacent tissue. Nevertheless, the underlying experiments were performed on monocultures. Further analyses using oral biofilm, which is known to be more pathogenic and resistant [7,8,9], are necessary and intended to reflect the realistic situation in vivo. The temperature of the implant and of the bone always remained under 40 °C. The application on aerobic, anaerobic, Gram-positive and Gram-negative bacteria led to a complete eradication of a bacterial load up to 6 log10 in the treated region. The size of the effective regions was always several times the diameter of the applicator nozzle. Even for a treatment time as short as 1 min, a complete eradication was observed over an area at least twice the size of the nozzle. Because the plasma can easily be seen, it serves as an important indicator for later clinical use to visualize and help navigate the area to be treated during a manual procedure. Thus, AmbiJet appears to be suitable for the prevention of periimplantitis, i.e., the inflammation of the tissue surrounding the dental implant. In addition, the electric field directs the plasma to the dental implant, which simplifies handling and helps to remove pathogenic biofilm in a controlled manner.

## Figures and Tables

**Figure 1 antibiotics-13-00735-f001:**
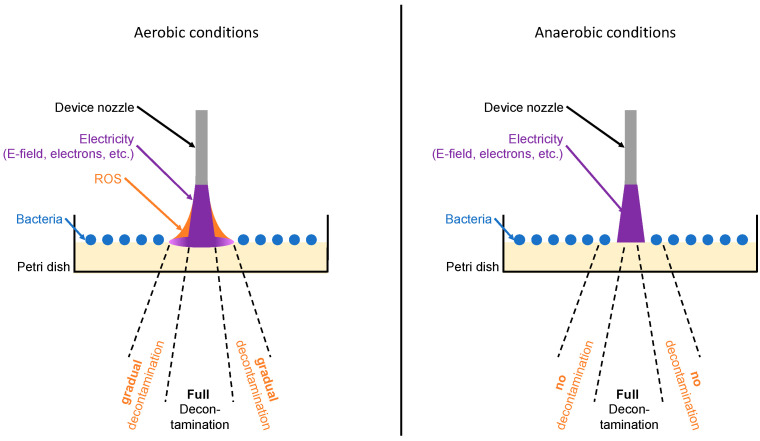
Scheme of plasma effect and influence of ROS under aerobic (**left**) vs. anaerobic conditions (**right**).

**Figure 2 antibiotics-13-00735-f002:**
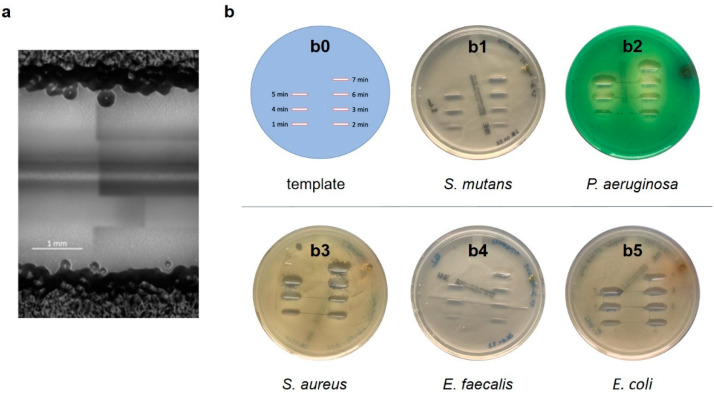
(**a**) Microscopic image showing the effective range of the plasma jet device applied to an *E. coli* bacterial lawn at a speed of 1 s/mm (4 runs). The dark area on the top and bottom represents the thick bacterial lawn. The light gray area in the center represents bacteria-free zone (3–4 mm); the inner line corresponds with the inner diameter of the nozzle. Scale bar: 1 mm. (**b**) Photographs of different bacterial lawns on agar plates after plasma application at a speed of 1 s/mm. (**b0**) Template showing treatment area for 1 to 7 min, (**b1**) *S. mutans*, (**b2**) *P. aeruginosa*, (**b3**) *S. aureus*, (**b4**) *E. faecalis*, (**b5**) *E. coli*.

**Figure 3 antibiotics-13-00735-f003:**
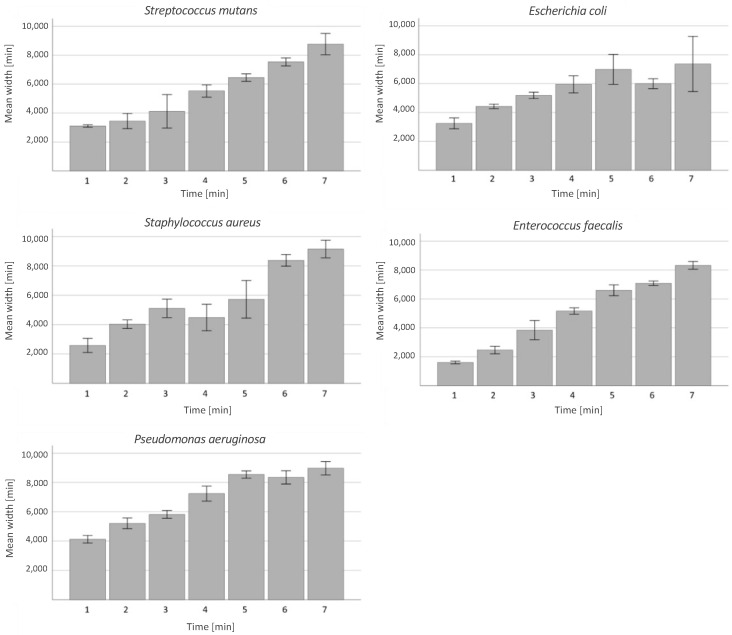
Width of the bacteria-free area with respect to the treatment time. Examined aerobic bacteria: *S. mutans*, *E. coli*, *E. faecalis*, *S. aureus* and *P. aeruginosa*. Chart indicated mean width with 95% confidence interval (CI) for treatment times of 1, 2, 3, 4, 5, 6 and 7 min.

**Figure 4 antibiotics-13-00735-f004:**
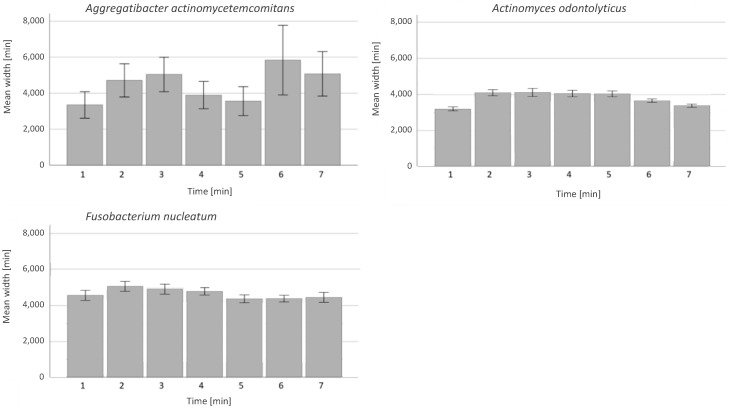
Time-dependent plasma effect on anaerobic bacteria. y-axis: bacteria-free area measured in width in µm. x-axis: examined bacteria from left to right: *A. actinomycetemcomitans*, *A. odontolyticus* and *F. nucleatum* with 95% CI; bars show treatment times of 1, 2, 3, 4, 5, 6 and 7 min.

**Figure 5 antibiotics-13-00735-f005:**
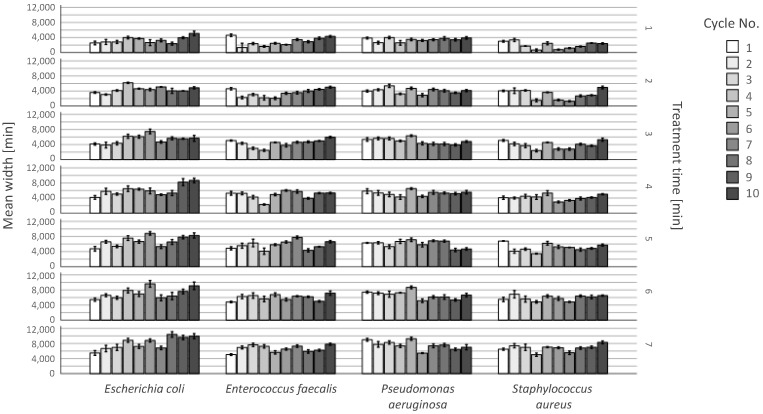
Long-term plasma effect on aerobic bacteria in an observation period of 50 days. Examined bacteria from left to right: *S. aureus*, *E. faecalis*, *E. coli*, *S. aureus* and *P. aeruginosa* with 95% CI. y-axis: bacteria-free area measured in width in µm. The bars show repeated treatment times of cycle 1, 2, 3, 4, 5, 6, 7, 8, 9 and 10. The rows represent the different treatment times.

**Figure 6 antibiotics-13-00735-f006:**
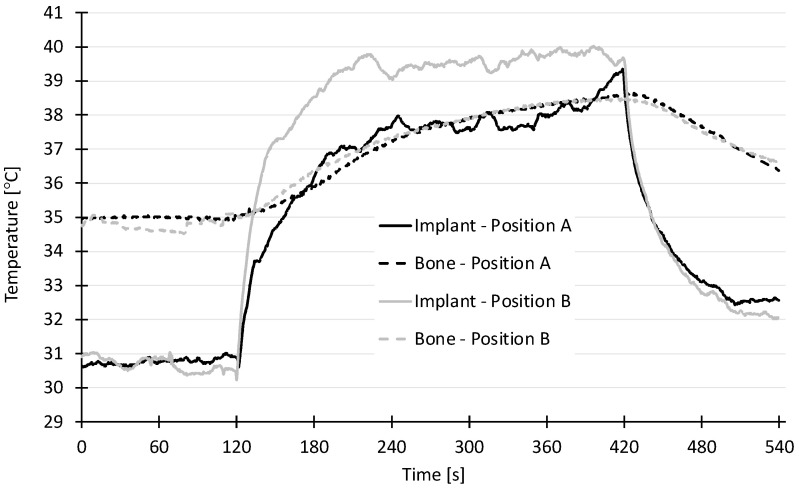
Temperature profile of the various sensors (implant position A and B, bone position A and B marked as lines), x-axis: time in s; y-axis: temperature in °C.

**Figure 7 antibiotics-13-00735-f007:**
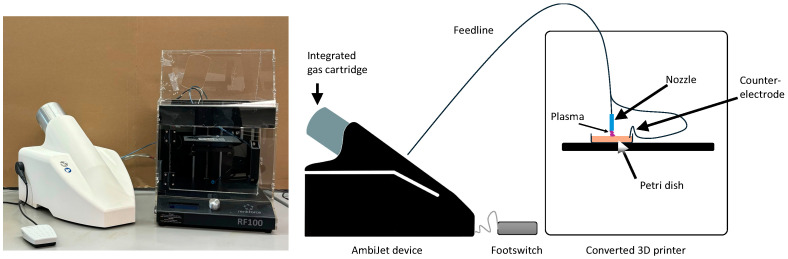
Experimental setup of the plasma. (**left**) Picture of the plasma device with integrated gas cartridge. (**right**) Schematics of setup. The nozzle comprises the powered electrode and the gas outlet. The counter-electrode is connected to the agar. The electrical field built between the powered electrode and the counter-electrode guides the plasma to the substrate. The footswitch is used to activate the plasma.

**Figure 8 antibiotics-13-00735-f008:**
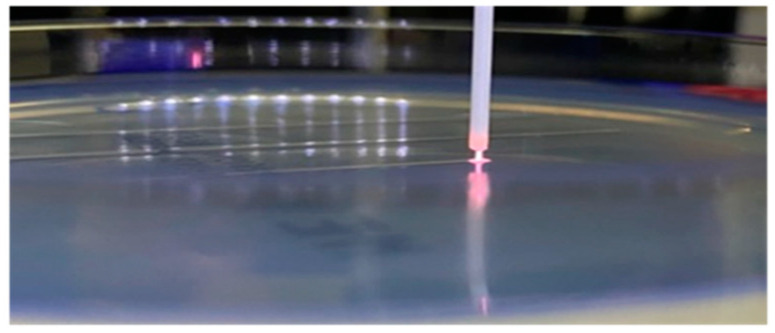
Nozzle and ignited plasma on the agar plate during plasma treatment in the customized 3D printer.

**Figure 9 antibiotics-13-00735-f009:**
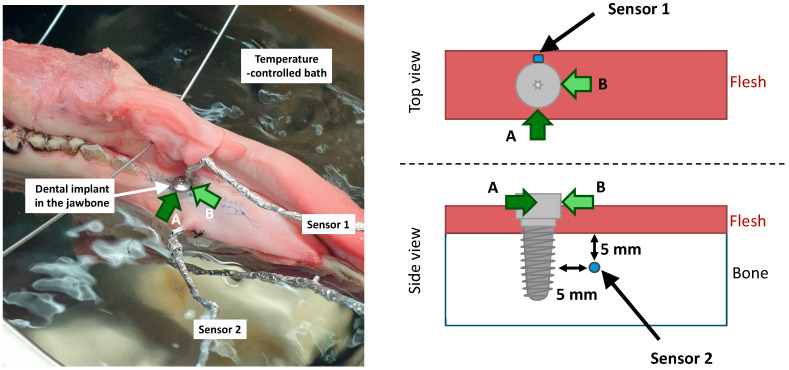
Temperature measurement setup displaying the sensors on and around the dental implant. (**Left**) picture of the implant inserted in the jawbone placed in a temperature-controlled bath. (**Right**) position of the temperature sensors. (Sensor 1 is located on the implant; sensor 2 is located in the bone 5 mm from the implant and 5 mm below the surface of the bone; a reference sensor measures the temperature of the bath; position A and B refer to the application spot of the plasma on the implant).

## Data Availability

All data presented in the paper are not openly available due to reasons of sensitivity. All data are available from the corresponding author upon reasonable request. All data are located in a controlled access data storage facility at the University of Freiburg.

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
