# Peer review of "Guided Plasma Application in Dentistry—An Alternative to Antibiotic Therapy"

_antibiotics, 2024, doi:10.3390/antibiotics13080735_

Round 1

Reviewer 1 Report

Comments and Suggestions for Authors

In general, this manuscript is well written. Please find my comments below.

 1. Line 184: …around 30% of the originally seeded bacteria alive.

Line 189: This selection procedure was repeated 10 times (10 cycles)…

My main question is how do the authors justify that choosing 30% and 10 cycles as key parameters can prove that plasma-treated bacteria do not develop resistance. Are there references for this? How were these numbers determined?

 2. In Figures 6 and 7, the numbers on the vertical coordinates are not readable. Please consider making the font larger and changing the units to millimeters.

Author Response

Response to the reviewer 1:

We thank you for the overall commendatory review of the manuscript. In our opinion, the argumentation in the revised manuscript has benefited greatly from the comments and suggestions by the reviewers. We have made every effort to implement all of the reviewers' suggestions. In the following pages, you will find a point-by-point response to each review report. We hope that the revised manuscript will meet your expectations.

Reviewer 1

Comment 1:

  1. Line 184: …around 30% of the originally seeded bacteria alive.

Line 189: This selection procedure was repeated 10 times (10 cycles)…

My main question is how do the authors justify that choosing 30% and 10 cycles as key parameters can prove that plasma-treated bacteria do not develop resistance. Are there references for this? How were these numbers determined?

Response 1:

Thank you for pointing this out. We agree that the justification must be included. Therefore, we adjusted the manuscript and included references in chapter “Discussion”, pages 8-9, lines 281-282, as well as in chapter “Materials & Methods”, page 10, lines 346-347.

Comment 2:

In Figures 6 and 7, the numbers on the vertical coordinates are not readable. Please consider making the font larger and changing the units to millimeters.

Response 2:

Thank you very much for your remark. We remodeled the figures accordingly, made the font larger, and stayed with units in µm for reasons of overview representation.

Reviewer 2 Report

Comments and Suggestions for Authors

The authors research on the use of  an application of plasma in dentistry as alternative to antibiotics therapy.  The work is general well designed and usefull for the use of this tool against bacteria. However there are several aspects that must be take into account before publication.

1) The authors only study the bactericidal effect on agar plates. however the study is intended to the treatment in dentistry. So this must be stated in the manuscript, because of the bacteria on implants could have biofilms and the resistance of the bacteria is variable in biofilms and is considerable higher than those on the agar.

introduction line 41. the authros put both comercial names as "listerine" together with specific chemical compound names as chorhexidine, please correct

Material and methods: what is the temperature of this cold plasma? this must be stated in the manuscript

Material and methods: line 139, the bacterial concentration is 106 bacteria per ml, but this was the concentration of the liquid of the tube used to contaminate the agar or is the concentration of bacteria in the agar itselef. Please add details on how was the bcterial load counted.

Material and methods  line 165 the authors stated a velocity of 60 mm/min. so the area treated durin 1 min were 6 cm?

Figure 6. how was measured the width of the bacterial. This should be stated into the methods part of the manuscript

lines 382-393. the authors must specify that the studies were carried out in agar surface, that is not the same that in biofilms, or in teeth directly.

Comments on the Quality of English Language

My perception is that the quality of english is good.

Author Response

Response to the reviewer 2:

We thank you for the overall commendatory review of the manuscript. In our opinion, the argumentation in the revised manuscript has benefited greatly from the comments and suggestions by the reviewers. We have made every effort to implement all of the reviewers' suggestions. In the following pages, you will find a point-by-point response to each review report. We hope that the revised manuscript will meet your expectations.

Reviewer 2

Comments 1:

The authors only study the bactericidal effect on agar plates. however the study is intended to the treatment in dentistry. So this must be stated in the manuscript, because of the bacteria on implants could have biofilms and the resistance of the bacteria is variable in biofilms and is considerable higher than those on the agar.

Response 1:

We appreciate this comment. Please see in chapter “Discussion”, page 12, lines 402-407 where we added this valuable input.

Comments 2:

introduction line 41. the authros put both comercial names as "listerine" together with specific chemical compound names as chorhexidine, please correct

Response 2:

We agree with the reviewer and changed it accordingly in chapter “Introduction”, page 1, lines 41-42.

Comment 3:

Material and methods: what is the temperature of this cold plasma? this must be stated in the manuscript

In agreement with the reviewer, the information about the temperature of the plasma was included in chapter “Materials & Methods”, page 9, lines 322-323.

Note: “Cold plasma” is a general term designing plasmas in which electrons and ions are not at the same temperature, i.e. do not possess on average the same velocity (non-equilibrium state or “non-thermal”). This is to differentiate from “hot plasma” such as lightnings or the sun. The temperature of the plasma jet itself is for most applications not relevant; its influence on the treated substrate is, however, essential.

Comment 4:

Material and methods: line 139, the bacterial concentration is 106 bacteria per ml, but this was the concentration of the liquid of the tube used to contaminate the agar or is the concentration of bacteria in the agar itselef. Please add details on how was the bcterial load counted.

Response 4:

The authors thank the reviewer for this question. An explanation on how the concentration of the bacteria had been determined was added in chapter “Materials & Methods”, page 9, lines 309-313 as recommended.

Comment 5:

Material and methods  line 165 the authors stated a velocity of 60 mm/min. so the area treated durin 1 min were 6 cm?

Response 5:

Thank you very much for your remark. This matter definitely needed clarification. Thus, we included the information in chapter “Materials & Methods”, page 10, line 337-339.

Comment 6:

Figure 6. how was measured the width of the bacterial. This should be stated into the methods part of the manuscript

Response 6:

Thank you for pointing this out. The width was measured using the software Fiji and a ruler for calibration. The information was added as recommended in chapter “Materials & Methods”, page 10, lines 346-347.

Comment 7:

lines 382-393. the authors must specify that the studies were carried out in agar surface, that is not the same that in biofilms, or in teeth directly.

Response 7:

The authors thank the reviewer for this valuable comment and completely agree that the difference of monocultured bacteria and an organized biofilm needs to be pointed out. We included this matter in chapter “Conclusions”, page 11, lines 391-394.